# Eradication of Yellow Crazy Ants, *Anoplolepis gracileps* Smith, from Lismore and Statistical Proof of Freedom Using Scenario Tree Analysis

**DOI:** 10.3390/insects16020117

**Published:** 2025-01-24

**Authors:** Robyn Henderson, Scott Charlton, Catherine Fraser, Barbara Moloney, Evan S. G. Sergeant, Bernard C. Dominiak

**Affiliations:** 1NSW Department of Primary Industries, The Ian Armstrong Building, 105 Prince Street, Orange, NSW 2800, Australia; robyn.henderson@dpi.nsw.gov.au (R.H.); scott.charlton@dpi.nsw.gov.au (S.C.); catherine.fraser@dpi.nsw.gov.au (C.F.);; 2AusVet Pty Ltd., Orange, NSW 2800, Australia; evansergeant@gmail.com

**Keywords:** incursion, threatening species, tramp ant, management

## Abstract

Exotic insects frequently invade countries, and many countries attempt to eradicate these exotic incursions. Regulatory authorities continue to monitor infested areas to understand whether the eradication methods were working and, finally, whether the eradication was successful. A common problem is how long should eradication and monitoring continue after the last invader is detected, because these are costly activities. We use the example of yellow crazy ant detection and eradication. Here, we used the statistical analysis of probabilities using scenario tree analysis to predict the likelihood of eradication based on different assumptions. This methodology has been used in both plant and animal kingdoms and will assist decision makers regarding when to cease activities.

## 1. Introduction

International trade in food and other commodities continues to increase to supply a growing world population [1]. Additionally, the tourism industry is key to the world’s economy. The dispersal of exotic species has been linked to both trade and tourism [2,3]. In Australia, all exotic pest detections and possible responses are coordinated by the National Biosecurity Consultative Committee (NBCC) and the Consultative Committee on Emergency Plant Pests (CCEPP) [4].

Invasive ants are a major exotic threat to Australia [5], especially in New South Wales (NSW), which is the most populous state. Argentine ants *Linepithema humile* (Mayr) were detected in NSW during the 1950s but eradication was unsuccessful [6]. Red imported fire ants *Solenopsis invicta* (Buren) were detected in Brisbane in 2001, and all Australian jurisdictions conducted surveillance as part of the CCEPP response to ensure the ant was not present in other states. No other exotic ants were detected in NSW between 2001 and 2006 [7,8,9]. However, a small colony of *S. invicta* was subsequently detected in Sydney and eradicated [10]. In November 2023, *S. invicta* were detected in northern NSW, and eradication is currently in progress.

Within the invasive ant group, yellow crazy ants (YCAs) *Anoplolepis gracileps* (Smith) are one of the six most widespread, abundant and damaging invasive ants globally [5,11]. The native range of YCAs is likely sub-Saharan Africa or Asia [5], but YCAs have a global distribution throughout the tropics and subtropics [12]. The recent history of invasions was reviewed, and the biology was summarised by Dominiak et al. [13]. YCAs have broad environmental, domestic and agricultural impacts usually causing considerable shifts in biodiversity [5,14,15,16]. The YCA is a general scavenger and predator of ground and arboreal creatures including arachnids, earthworms, insects, small isopods, myriapods and molluscs [14,17]. On Christmas Island, YCAs caused a decline in the populations of the red crab (*Gecarcoidea natalis* Pocock), adversely impacting the forest ecosystem [17,18]. The red crabs play a pivotal role as they reside in forests eating the leaf litter on the forest floor [19], and the declining red crab populations resulted in an accumulation of litter. The crab’s activities influence nutrient turnover and the survival of tree seedlings [18,19]. The YCAs tended the lac scale (*Tachardina aurantiaca* Cockerell) seeking sugars (such as honeydew), leading to massive increases in scale populations. In turn, this resulted in subsequent physiological stress for trees and a sooty mould infestation resulting from the widespread honeydew “rain” from scale outbreaks [17,18]. Additionally, YCAs adversely impacted burrow-nesting sea birds on Christmas Island [19,20]. The Christmas Island example demonstrates how YCAs can dramatically and adversely impact ecosystem dynamics.

In Australia, the YCA was first detected in northeast Arnhem Land in 1982 and spread across 2500 km^2^ [16,21]. In eastern Australia, the YCA was detected in several locations in Queensland [13]. Further south in eastern Australia, YCAs were detected at Goodwood Island, Yamba, in north coast NSW and eradicated [13]. Statistical proof of eradication was required and estimated using scenario tree analysis. The scenario tree analysis uses stochastic models to describe each component of a surveillance system (SSC) and was used to estimate the sensitivity of each SSC. Martin et al. [22] described the process of building and analysing the models and the techniques used to take into account any lack of independence between units at different levels within the SSC. The combination of sensitivity estimates from multiple SSCs into a single estimate is used to predict the probability of freedom [22].

Originally, scenario tree analysis was developed to demonstrate the absence of animal diseases [22] and, subsequently, was used to declare NSW free from equine influenza [23]. Later, scenario tree analysis was developed for plant health programs. Scenario tree analysis was used to estimate the probability of eradication of YCAs at Yamba [13]. Similarly, *S. invicta* were detected in Sydney, the capital city of NSW, and eradicated under CCEPP funding arrangements. Scenario tree analysis was used to statistically demonstrate freedom [10]. Apart from ants, scenario tree analysis has been used to demonstrate the eradication of Fiji disease from central Queensland [24]. Here, we review the period between detection in May 2018 and December 2019 of *A. gracileps* at Lismore in northern NSW and the response program. Finally, we used scenario tree analysis to demonstrate at a certain probability of freedom that the eradication program was successful.

## 2. Materials and Methods

### 2.1. Detections

The detection and response are described briefly to provide perspective to the eradication assessment. On 14 May 2018, a member of the public reported a suspected YCA in the Lismore central business district (CBD). A sample was collected and sent to Orange Agricultural Institute, and YCA was confirmed by Dr. Ainsley Seago on 21 May 2018. The Australian Government was notified on 21 May as part of the functions of the CCEPP (see [4] for details). A local control centre was established by the NSW Department of Primary Industries (DPI) on 4 June to assess the extent of the infestation. Genetic analysis (DNA testing) subsequently confirmed the ants were YCAs and likely originated from Queensland and not from the earlier detection at Yamba [13]. Sites were confirmed as infested throughout the CBD and, subsequently, in the Terania Creek area—approximately 21 km from the Lismore CBD.

### 2.2. Response

An incident action plan was developed by the NSW DPI. The North Coast Local Land Services and the NSW DPI led a response assisted by the National Parks and Wildlife Service, Rous County Council, Lismore City Council and NSW DPI Cattle Tick Inspection staff based at Wollongbar. Also, Landcare, NSW Rural Fire Service, State Emergency Services and other community volunteers contributed to the program through surveillance activities. The response plan primarily focused on the treatment of the CBD of Lismore, the establishment of pest control zones, surveillance for the detection of additional YCAs (if present) and the pre-emptive treatment of high-risk areas to further reduce the risk of establishment or spread of additional nests.

The Lismore infestation was restricted to a concentrated 19 ha (including buildings) area within the Lismore CBD, and the YCAs were very abundant within this area. The second infestation at Terania Creek is believed to be linked to building materials moved from the CBD. The Terania Creek infestation was very problematic because of the difficult terrain and dense vegetation at that location. This infestation was of particular concern because of its immediate proximity to Nightcap National Park—a significant World Heritage area.

Surveillance activities, including trapping, luring, odour detection dogs and visual detection, commenced on 21 May 2018 and ceased on 12 October 2019. Lures were used during the delineation process. These consisted of either jam (sugar based) or fish-based cat food (protein based) with no pesticide. In the simplest form, lures were placed on trays and reinspected two hours later, and the YCA population was assessed visually. Alternatively, lures were placed on a sticky mat and reinspected after 24 h to assess the population. Visual detection was conducted by the program staff, looking for ants in places they were likely to inhabit.

The DPI engaged a local consultant to re-train their koala odour detection dog “Jet” to detect YCAs. Jet is the first YCA detection dog in Australia and has subsequently been deployed in Arnhem Land to assist with another program. The use of the detection dog provides a very high level of confidence of presence/absence, as the dog does not exhibit the same biases as humans. A dog’s nose can detect scent at least 1000 times better than a human can [25]. A GPS collar was attached to the dog to map the areas inspected. Any sites with suspect ants were marked for subsequent inspection and possible treatment. Several visits (September to November 2019) by the odour detector dog did not detect any ants at either site since May 2019.

There was considerable public consultation. By 29 June 2018, there were 107 submissions received from the public via the Ant Report Call Centre and via email. An additional 427 contacts were received by 14 December 2018. A schools awareness program was established to enhance awareness and to potentially assist with ant collection or reporting. A general biosecurity direction, under the NSW Biosecurity Act 2015, was issued to occupants at all infested sites to prohibit the movement of risky material including garden vegetation, timber, soil products, gravel and machinery.

Extensive active surveillance was undertaken during 2018 to 2019 in accordance with the response plan. Additionally, a passive surveillance program was put in place during this time. The DPI conducted an awareness campaign in the region, resulting in several hundred public reports. These reports were consistent with the known distribution of the ants. Overall, there were 245 case reports investigated. Movement controls for soil and associated products were in place during this period, and baiting within a 500-m perimeter of the original nest was undertaken in accordance with the response plan. The total response costs were about AUD 500,000.

### 2.3. Pesticides

At different times of the year, ants seek sugar or protein depending largely on the season. The Australian Pesticides and Veterinary Medicines Authority (APVMA) approved off-label permits to use pesticides not specifically registered for YCAs. Most treatments were products containing the active ingredient fipronil, at different rates and formulations. At many sites, the protein-based bait Antoff^®^ (containing 0.01 g/kg fipronil) was used according to the permitted off-label use (PER86559—31 May 2018 to 31 May 2021), which included s-methoprene. Primarily, this was a broadcast bait intended to Thank restrict population growth. Additionally, a permit (PER87623—17 January 2019 to 31 January 2020) was approved for 100 g/L fipronil bait for undeveloped bush and non-crop situations. Subsequently, PER87750 (26 March 2019 to 31 March 2022) permitted 100 g/L fipronil to be used in residential, commercial and industrial areas in which YCA populations were the highest. The bait product’s use was permitted up to 15 kg hydrogel beads per hectare. This preparation consisted of 1.3 mL of Termidor^®^ added to 6 kg sugar, 14 L of water and 0.153 kg hydrogel crystals, with an equivalent rate up to 0.096 g fipronil per hectare. All permits detailed the limitations of use such as proximity to watercourses. Recently, PER87750 was extended to 31 March 2026 to be ready for any future YCA detection. Baits were used as a broadcast bait to restrict the population. The same formulation was used in “bait traps”, which were traps that were reinspected after several days. Assessment of the YCA population was performed visually or by assessing the amount of bait removed from traps.

The initial baiting of 80 Ha (CBD and surrounding buffer) was completed by 10 June 2018 and used 36 kg (Antoff^®^ in 274 bait traps (Figure 1 and Figure 2)) across 196 broadcast lines. At Terania Creek, 24 kgs (Antoff^®^) were broadcast. The YCAs in the CBD had a preference for carbohydrates over protein and showed little interest in the protein-based Antoff^®^ bait. However, Antoff^®^ was consumed at Terania Creek. Goonellabah (an eastern suburb of Lismore) was treated with preventative baiting on 13 June 2018. Baiting operations were repeated at three monthly intervals. During the 2018/2019 summer, bait longevity was limited because of frequent wet weather; the baits became mouldy, and the ants seemed less attracted to these baits. The fipronil/sugar-water/hydrogel bead bait (720 kg) was used at Terania Creek on 16 March 2019. There were four rounds of the fipronil bait, and Antoff ^®^ was used at the Lismore CBD by 7 April 2019. At the infested sites, no additional YCAs were detected at Lismore CBD or Terania Creek after May 2019.

## 3. Statistical Analysis for YCA Proof of Freedom

An active surveillance program of 600 properties and passive surveillance, underpinned by an intensive communication strategy for residents, was undertaken to provide evidence for proof of freedom for the presence of YCAs in the Lismore region of NSW including Terania Creek. The surveillance results were analysed statistically to provide evidence for proof of freedom from YCAs. The unit of interest (population at risk) for the analysis was the number of households/dwellings within the 10 km radius of the YCA incursion zone. The proof of freedom was assessed by the probability that ants were not present at specified levels, given that no nests were found.

### 3.1. Methods

We developed a stochastic simulation model to evaluate the effectiveness of both passive and active YCA surveillance undertaken in a 10 km zone around the two incursion sites and the treatment area at Lismore, NSW. The model was developed in Microsoft Excel, using the PopTools add-in. The model was a risk-based statistical model (Monte Carlo simulation) of the surveillance undertaken, with the model outputs being as follows:
(1)The surveillance sensitivity (probability of detecting one, two or five nests) for each surveillance event and a combined sensitivity.(2)The probability of freedom (probability that ants were not present at the design prevalence of one, two or five nests), given that no nests were found.

Key model inputs were entered as probability distributions, and the model was run for 10,000 iterations, producing probability distributions for model outputs. This methodology has been used widely within the animal health area to demonstrate freedom from disease [13]. It was used in 2015 for red imported fire ant surveillance evaluation for proof of freedom [10]. The model notionally divides the surveillance area into a hypothetical “population” of equal-sized units. A hypothetical number of infested units (the design prevalence) is assumed (in this case one, two or five nests). The model estimates the probability that one or more of these infested units would be detected by the surveillance undertaken. The model inputs are summarised in Table 1 and described in more detail below.

### 3.2. Design Prevalence

The analysis was repeated for three separate design prevalence values of one, two and five nests within the 10 km zone, representing approximately 0.007%, 0.015% and 0.037% of the units infested, respectively. Passive surveillance of residential properties (owner notification of suspected cases) was included and combined with the active surveillance survey in which 600 randomly selected and inspected properties were included.

### 3.3. Prior Probability of Freedom

We required a prior value (the level of confidence of freedom prior to the surveillance being undertaken) to calculate the probability of freedom arising from a surveillance activity. For the survey, a value of 0.5 was assumed, indicating a position of no knowledge and that the presence or absence of additional nests is equally likely. The probability of further introduction was determined to be zero.

### 3.4. Number of Susceptible Units

Within a 10 km radius of the YCA incursion, excluding the treatment area, there were a total number of 13,342 unique households, which may have been infested with a YCA nest. The passive surveillance of residential properties (owner notification of suspected cases) was combined with an active survey in which 600 randomly selected properties (Figure 3) were included from within the 10 km radius. The relative importance of each surveillance component in the overall system’s sensitivity and proof of freedom is demonstrated in Figure 4 for the three design prevalence levels.

### 3.5. Active Surveillance

These 600 properties were randomly selected and had surveillance activities undertaken on the dwelling/household by the NSW DPI staff and one dog trained in the detection of YCA nests. The number of 600 for the active targeted survey was determined to achieve 95% confidence of detecting one nest at a design prevalence of 0.5% with an assumed specificity of 100% (a positive YCA detection is 100% specific). Of these households, 2826 were on an area greater than or equal to 0.25 ha, and 10,516 households were on an area less than 0.25 ha. The sample of 600 households was randomly allocated using the Geospatial Modelling Environment [26] and included 139 households ≥0.25 ha and 461 households <0.25 ha (Figure 3).

### 3.6. Passive Surveillance

An extensive communication strategy was undertaken in the region, which included local media, information stands at local events, social media, web information, information documents in collaboration with local councils and school outreach. This was to build a high level of awareness and to account for the passage of time for owners to become aware of a nest on their property. The probability that an owner will notice a nest, the probability that the owner will notify the suspected nest and the probability that the report will be followed up and investigated were modelled as PERT distributions (smoothed triangular distributions described by minimum, most likely and maximum values), as shown in Table 2. The probability that a nest will be confirmed, if present and notified, is assumed to be 1.

### 3.7. Other Inputs—“Test” Sensitivities

The sensitivity of a test is the probability of a positive test result given that the unit tested is infested. In this case, the unit of interest is the spatial unit defined previously (the household or dwelling), and the “test” is one of several options used during the surveillance. The sensitivity of passive surveillance is based on the combined conditional probabilities that an owner will notice a nest, that they will then notify the relevant authorities about the presence of the nest, that the authorities will investigate the notification appropriately and that it will be confirmed as a YCA nest.

## 4. Results

The overall system sensitivity of detecting YCA nests and the probability of freedom for YCAs is detailed in Table 3 (if communication strategies were used) and Table 4 (if communication strategies were not used).

### 4.1. Surveillance Sensitivity

The mean surveillance sensitivity for the active survey ranged from 4.7% (five nests) to 0.9% (one nest), which is not very sensitive by itself. When this was combined with the passive surveillance sensitivity, the overall system sensitivity was at 57.4% (40.4–76.0) for one nest, 80.9% (64.8–94.1) for two nests and 97.9% (92.5–99.9) for five nests.

### 4.2. Probability of Freedom

We combined the results of the two surveillance systems undertaken and that provided a 70.4% (62.7–80.7) probability of freedom of detecting one nest, an 84.4% (73.9–94.4) probability of freedom for two nests and a 98% (93.1–99.9) probability of freedom for five nests. If spread from the initial incursion is assumed or that the YCAs are endemic in the area or that there have been multiple incursions, then it is more likely that multiple nests will be present. Therefore, the estimates assuming two or five nests would be more appropriate than that for one nest.

## 5. Discussion

It is difficult to determine conclusively how the ants arrived at Lismore. However, we suspected that the YCAs arrived either in consignments of tractor parts, in imported timber or via equipment used by a Queensland-based company.

All the sites received multiple rounds of the fipronil insecticide bait, following protocols developed by other national YCA programs. The baiting program commenced using the commercially available Ant-off^®^ bait but later adopted an experimental fipronil-based sugar and water crystal bait when YCAs showed an aversion to the protein-based products. The water crystal bait proved to be extremely effective, with good knock down within days of application. Fipronil caused negligible adverse effects to non-target arthropods and vertebrate fauna on Christmas Island and is appropriate for YCA control [18]. Currently in Australia, there are ten permits to control YCAs, eight of which use fipronil. In the past, other compounds included S-methoprene, pyriproxyfen and hydramethylnon (Kidston, pers. comm., 23 January 2024) were available.

The size of the Lismore YCA population may have been influenced by several factors. NSW, including Lismore, was extensively surveyed as part of the response to red imported fire ant, *S. invicta*, detections in Queensland [8]. All ants were sent for identification, and no exotic ants were detected. After 9558 site inspections, no YCAs were detected up to 2006 [9], so the incursion is likely to have happened after 2006.

YCA colony expansion by budding alone ranged from 37 to 402 m y^−1^, so colony expansion is relatively limited [5]. Ant species known to be resident in the region included *Calomyrmex* sp., *Cardiocondyla* sp., *Leptogenys* sp., *Pheidole* sp. and *Plagiolepis* sp. [9], and these resident ants could have slowed YCA colony expansion [27]. Conversely, the YCAs coexisted with ants <2.5 mm [28]. YCAs reach high abundance when accessing honeydew from sap-sucking insects [29], but these insects may not have been abundant at Lismore, particularly in the CBD. In northern Australia, YCA nests are not randomly distributed and are frequently located at the base of a tree, particularly large trees. Phloem-feeding insects provide important carbohydrate (honeydew) resources for ants. Carbohydrates dominate *A. gracilipes* colony dietary requirements and regulate population density [30]. Baits in a sugar-rich carrier were more attractive to the YCAs than protein-based baits [31]. In the Lismore CBD, we speculate that there were other sources of carbohydrate (such as food waste) to support colony establishment and expansion.

The increase in the infested area and the number of nests were highly correlated with time and increased exponentially after about 1000 days [30]. Early detection is key to a successful eradication before colony size can increase exponentially. Therefore, a public awareness program regarding exotic pests is critical to detection and eradication.

Based on the results of our scenario tree analysis, we suggest that there was a high level of confidence that the YCAs were successfully eradicated from the Lismore region in a relatively short timeframe. This confidence seemed to be confirmed by subsequent monitoring. Additionally, the results demonstrate that passive surveillance, supported by high levels of community awareness, provides a highly effective tool for confirming freedom from pests with a high local impact, such as YCAs.

There have been other eradications of YCAs. Lismore is about 80 km north of Goodwood Island, where a YCA colony was eradicated in 2006 and declared YCA free in 2008 [13]. In northern Australia, 22 discrete infestations over 79 ha were eradicated. Additionally, a further ten locations over >500 ha was presumed eradicated [30]. In Queensland, YCAs were eradicated in 2005 from 6 ha at Portsmith, Cairns. Similarly, YCAs were eradicated from 6.5 ha at Woree, Cairns, in 2006 [32]. Conversely, YCAs established and expanded on Christmas Island for 50 years before they were detected [11]. Aerial baiting with fipronil is being used on Christmas Island to minimise YCA populations [18].

### Post-Assessment Period

For Lismore, YCA freedom was notionally accepted on 19 December 2019, and surveillance ceased at this time. On 25 February 2021, a YCA was reported by a member of the public, in the Lismore CBD and tested genetically. The test indicated that the 2021 detection originated from Brisbane, as did those collected from Lismore during 2018 and 2019. The sniffer dog was redeployed and found a second infested premises (25 March 2021), in a tree five metres from the February 2021 detection. Further inspections found a total of 13 infested premises within a 300 m radius. Treatment recommenced on 26 March. Baits and lures were placed in April and July 2021.

In February 2022, a severe rain event struck the northern rivers region of New South Wales [33]. The Lismore CBD was submerged by many metres of water, and a disaster was declared. This flood was followed by two other flooding events. After the initial clean-up of massive flood damage, there was another round of treatment and surveillance in November and December 2022. Lismore is currently in disaster recovery mode at the time of writing this manuscript.

Our analysis has some lessons for future similar circumstances. Our model predicted a 70.4% and 84.4% chance of detecting a one- or two-nest scenario, respectively. Conversely, there was a 26.9% chance of not detecting one nest and a 15.6% chance of not detecting two nests. We did subsequently detect a likely surviving population, and we suggest the subsequent detection was consistent with the model’s predicted probabilities. When undertaking the analysis, the probability of someone noticing a nest (0.5, 0.75, 0.95) and then reporting it (0.5, 0.8, 0.99) were the minimum/most-likely/maximum probabilities that we used in the model. These probabilities need to be revisited, given the subsequent events in November 2021.

## 6. Conclusions

Urban areas, particularly CBDs, may present more challenges for detection than agricultural or residential areas. Our CBD surveillance was conducted along areas easily accessible by humans and dogs. We guess that very small numbers of a YCA population survived treatment, possibly as unhatched pupae and evaded detection during the time it took to build up a larger population, going unnoticed by tenants and not active above ground during subsequent sniffer dog surveillances. For future eradications of similar ants in urban or CBD areas, we propose that the model needs to be run with a higher detection probability, which would lead to a higher sensitivity for one or two nests. However, we do not know what that level might be. Any eradication and subsequent surveillance are a balance between high cost and potentially unnecessary surveillance and treatment costs, compared to achieving eradication.

## Figures and Tables

**Figure 1 insects-16-00117-f001:**
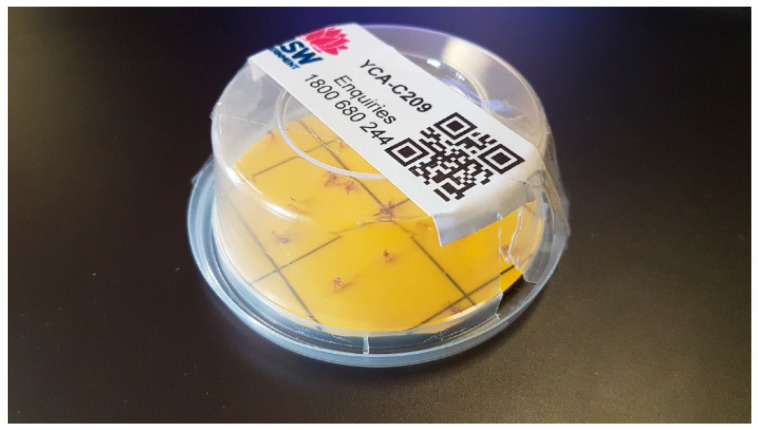
Yellow crazy ant lure monitoring trap with a sticky mat. The entry point is at the bottom right, and the bait is on the far side of the entry point. YCAs can be seen stuck on the sticky mat. Lure traps were inspected after 24 h.

**Figure 2 insects-16-00117-f002:**
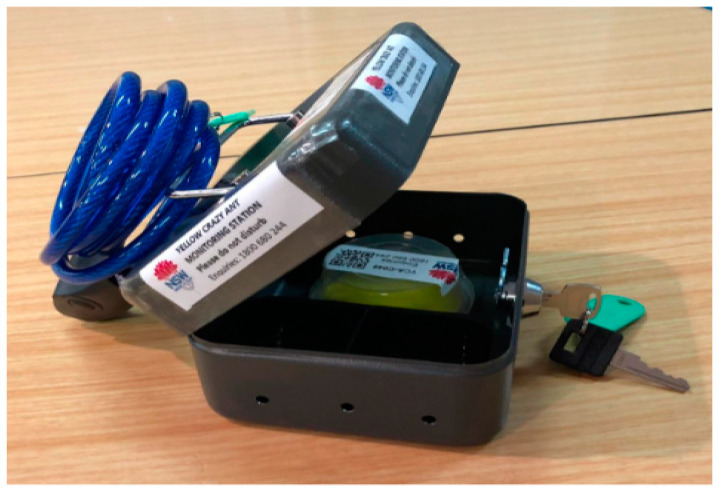
Secure monitoring lure bait station to prevent theft or removal. The metal box protected the lure trap but allowed foraging ants to enter. The lockable cable prevented unauthorised removal, particularly in public places. The same device was used for bait trapping.

**Figure 3 insects-16-00117-f003:**
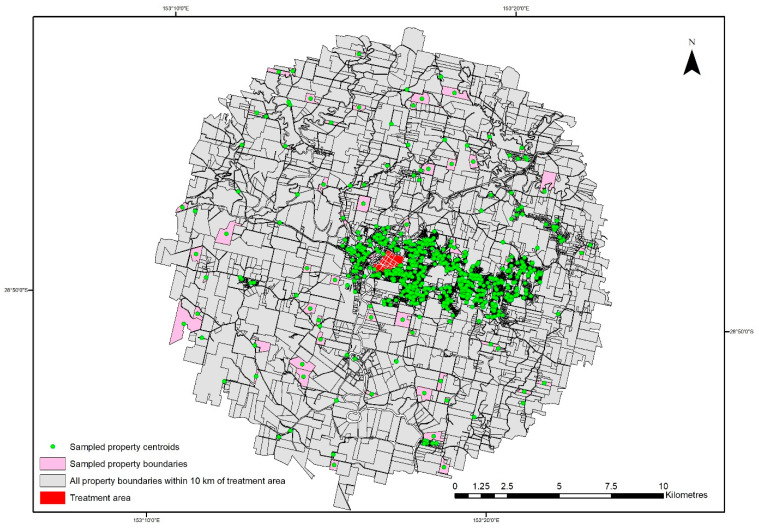
Spatial distribution of 600 sampled properties within the 10 km boundary of the treatment area (Lismore CBD).

**Figure 4 insects-16-00117-f004:**
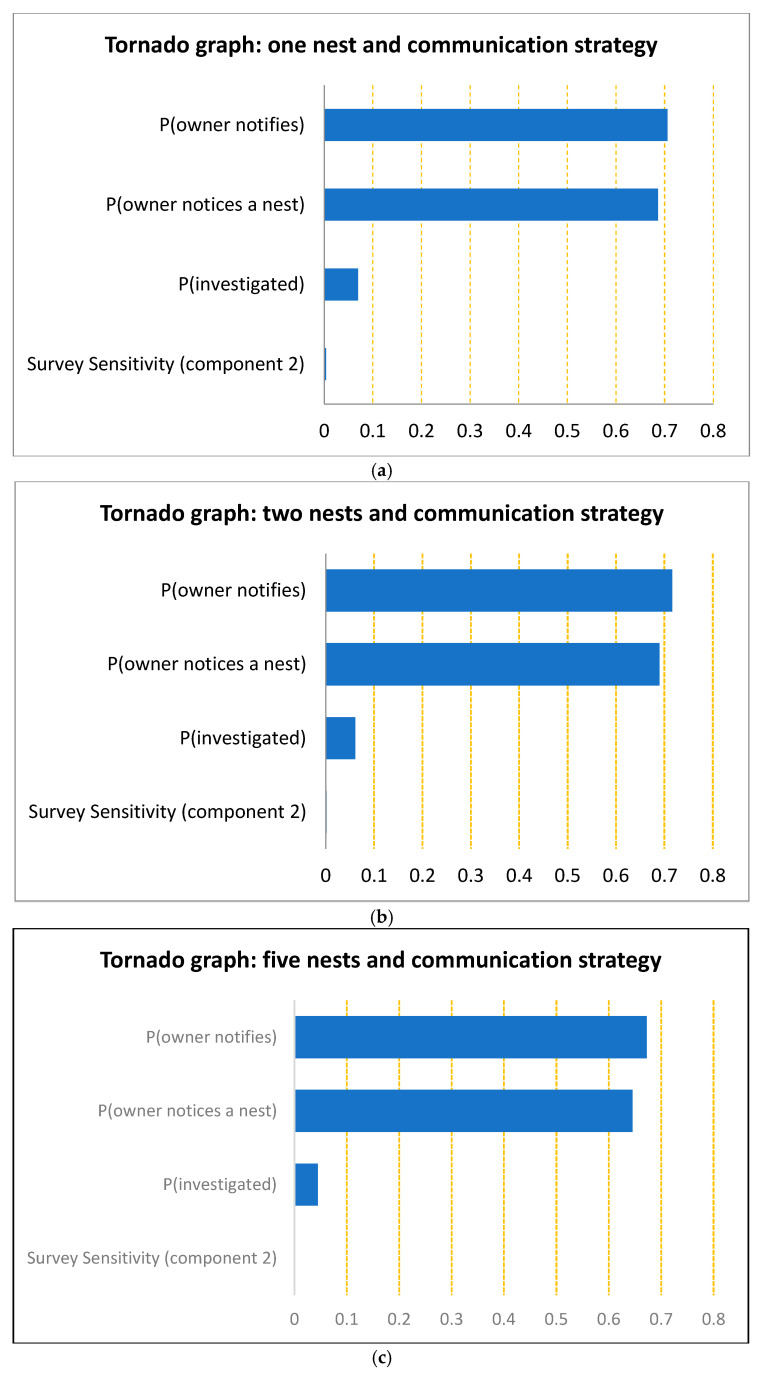
The relative importance of each surveillance component in providing the overall system sensitivity and proof of freedom is shown in the tornado graphs. The sensitivity levels are (**a**) one nest, (**b**) two nests and (**c**) five nests.

**Table 1 insects-16-00117-t001:** Input values for risk-based model to evaluate surveillance data for YCA proof of freedom.

Inputs	Values		
Design prevalence			
Number of nests	1	2	5
Prior probability of freedom	0.5		
Probability of introduction	0		
Number of “units”	13,342		
Survey sample size	600		
Design prevalence (%)	0.007%	0.015%	0.037%

**Table 2 insects-16-00117-t002:** Inputs for test sensitivities for active and passive surveillance for YCAs.

Test Sensitivities	Distribution	Min	Most Likely	Max
Active surveillance				
Random survey	PERT	0.4	0.5	0.6
Passive surveillance				
P (owner notices a nest)	PERT	0.5	0.75	0.95
P (owner notifies)	PERT	0.5	0.8	0.99
P (investigated)	PERT	0.95	0.99	1
P (confirmed)	PERT		1	

**Table 3 insects-16-00117-t003:** Likelihood of freedom if communications strategies were used and included in the analysis.

With Communication Strategy	5 Nests		2 Nests		1 Nest	
	Mean	95% PI	Mean	95% PI	Mean	95% PI
Active survey sensitivity	0.047	0.026–0.068	0.019	0.011–0.028	0.009	0.005–0.014
Passive surveillance sensitivity	0.978	0.920–0.999	0.806	0.639–0.940	0.569	0.396–0.759
Overall system sensitivity	0.979	0.925–0.999	0.809	0.648–0.941	0.574	0.404–0.760
Probability of freedom	0.98	0.931–0.999	0.844	0.739–0.944	0.704	0.627–0.807

**Table 4 insects-16-00117-t004:** Likelihood of freedom if communications strategies were not used or excluded from the analysis.

No Communication Strategy	5 Nests		2 Nests		1 Nest	
	Mean	95% PI	Mean	95% PI	Mean	95% PI
Active survey sensitivity	0.0698	0.049–0.087	0.028	0.020–0.036	0.014	0.010–0.018
Passive surveillance sensitivity	0.874	0.745–0.978	0.58	0.421–0.782	0.356	0.240–0.532
Overall system sensitivity	0.883	0.766–0.979	0.593	0.441–0.787	0.366	0.253–0.538
Probability of freedom	0.898	0.810–0.979	0.714	0.642–0.824	0.613	0.572–0.684

## Data Availability

Data will be available on request.

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
