# Peer review of "Eradication of Yellow Crazy Ants, Anoplolepis gracileps Smith, from Lismore and Statistical Proof of Freedom Using Scenario Tree Analysis"

_insects, 2025, doi:10.3390/insects16020117_

Round 1

Reviewer 1 Report

Comments and Suggestions for Authors

This paper describes the response to an infestation of yellow crazy ants in the Lismore area of New South Wales. Taken as a whole, the paper is interesting and provides useful information for people with an interest in yellow crazy ant invasion or land managers looking to deal with a similar invasion. Some more detail could be added in places, especially regarding specifics of the baits used and the surveillance methods.

L22: Pantropical might fit better than global – YCA distribution is largely limited to the tropics and subtropics.

L26: Spelling error – ‘ign’

L42: The tense of the word ‘was’ doesn’t fit with the rest of the paragraph. Suggest using ‘has been’. Dispersal is still linked with both trade and tourism.

L50: Not required, but this paragraph could potentially be updated to reflect recent incursions of RIFA into northern NSW.

L56: this sentence appears truncated and would be clearer if rewritten.

L103: it might be good to mention the Queensland infestation in the intro, seeing as it is mentioned here.

L151: I was under the impression that Antoff did not have a registered label until this year (2024).

L154: Suggest checking that 100g fipronil /L is correct. This is an incredibly high rate of application for fipronil if that is what is actually applied on-ground.

One hundred g/L is the concentration of Termidor concentrate. I am assuming that this was the source of fipronil? If yes, there should be more detail added describing the use of Termidor and the final concentrations or rate of fipronil in bait that was actually applied on ground, which will hopefully have been at a significantly lower concentration than 100g/L. What % sugar water used for bait, as described on L169? Was sucrose used, or different sugar?

L162: More discussion of the monitoring trap shown in Figure 1 would be good. It’s not clear why a monitoring trap (shown in fig1) was being loaded with bait (as suggested on L162).

L163: This is interesting, given that Antoff baits are so successful elsewhere in Australia. Can more details be provided of the methods used to determine macronutrient preference?

L188: I am not qualified to adequately review this method of determining proof of freedom. This section needs to be reviewed by a statistician.

L316: These alternative compounds are still available (change tense).

L320: Were these RIFA surveys also looking for YCA?

L323: It is very speculative to suggest that these species might have any influence on YCA spread rates. Is there any evidence in the literature that these particular species can slow YCA spread? It is difficult to see how the presence of any of these genera, would have any influence on YCA given their very different natural histories. This sentence also runs counter to your sentence on L325 as most Cardiocondyla and Plagiolepis, and many Pheidole, are sub 2.5mm.

Author Response

Reviewer 1

This paper describes the response to an infestation of yellow crazy ants in the Lismore area of New South Wales. Taken as a whole, the paper is interesting and provides useful information for people with an interest in yellow crazy ant invasion or land managers looking to deal with a similar invasion. Some more detail could be added in places, especially regarding specifics of the baits used and the surveillance methods.

L22: Pantropical might fit better than global – YCA distribution is largely limited to the tropics and subtropics. Text adopted

L26: Spelling error – ‘ign’ – text corrected

L42: The tense of the word ‘was’ doesn’t fit with the rest of the paragraph. Suggest using ‘has been’. Dispersal is still linked with both trade and tourism. text corrected

L50: Not required, but this paragraph could potentially be updated to reflect recent incursions of RIFA into northern NSW. Text amended

L56: this sentence appears truncated and would be clearer if rewritten. Text amended

L103: it might be good to mention the Queensland infestation in the intro, seeing as it is mentioned here. Text added to the introduction

L151: I was under the impression that Antoff did not have a registered label until this year (2024). The reviewer is correct – text amended

L154: Suggest checking that 100g fipronil /L is correct. This is the rate taken directly from the Permit instructions. This is an incredibly high rate of application for fipronil if that is what is actually applied on-ground. Text added to explain the formulation.

One hundred g/L is the concentration of Termidor concentrate yes, it was. I am assuming that this was the source of fipronil? If yes, there should be more detail added describing the use of Termidor and the final concentrations or rate of fipronil in bait that was actually applied on ground, which will hopefully have been at a significantly lower concentration than 100g/L. additional text added What % sugar water used for bait, as described on L169? The formula for the bait is added into the text Was sucrose used ordinary white sugar was used, or different sugar? Text added to explain the formulation.

L162: More discussion of the monitoring trap shown in Figure 1 would be good. It’s not clear why a monitoring trap (shown in fig1) was being loaded with bait (as suggested on L162). The traps need a bait to draw the ants into the trap and the program called these as “bait traps”. This makes a distinction from monitoring traps for fruit flies which use a pheromone lure and clearly these are not bait traps. Text added to clarify these points.

L163: This is interesting, given that Antoff baits are so successful elsewhere in Australia. Can more details be provided of the methods used to determine macronutrient preference? I added text indicating that ants require sugar or protein at different seasons of the year.

L188: I am not qualified to adequately review this method of determining proof of freedom. This section needs to be reviewed by a statistician. Scenario tree analysis is widely used and accepted in biosecurity and emergency response work areas. I would be happy for a statistician to review the statistical methods, as long as they are familiar with this or similar techniques (eg Bayes net).

L316: These alternative compounds are still available (change tense). They were available at the time of writing the manuscript.

L320: Were these RIFA surveys also looking for YCA? Text added

L323: It is very speculative to suggest that these species might have any influence on YCA spread rates Yes, this is speculation but any exotic has to establish a self sustaining colony while combating current occupiers. The same applied for human colonization. This is not a new concept. A reference was added. Is there any evidence in the literature that these particular species can slow YCA spread? This is a general concept and there are several references – I added one reference. It is difficult to see how the presence of any of these genera, would have any influence on YCA given their very different natural histories perhaps so, however in the initial stages, any ant colony has few reserves and a low propagule population. Endemic ants are likely to have adequate housing, food sources and stored reserves, and a high population. The theory has some merit, purely based on logistics and resources. This sentence also runs counter to your sentence on L325 as most Cardiocondyla and Plagiolepis, and many Pheidole, are sub 2.5mm. Perhaps so, we are proposing some speculation and it is the role of scientists to explore all ideas, not just to document what is known.

Reviewer 2

I have made edits (grammar) and comments and inquiries (that must be addressed) in pencil on the MS for the author's consideration. Thanks for these comments. I will number each comment and respond where I can read the handwritten. The MS is well-written thank you, but in area some things are not explained or defined (methods) or left unmentioned.

The study would have greatly benefited by having an entomologist/myrmecologist involved, esp one experienced in ant management This is an epidemiology paper using ants as the example. It is not primarily an entomology paper. Ant control is quite difficult, and consultation with an experienced entomologist might have highlighted that fact that large ant populations (tawny crazy ant, Argentine ant) are not eliminated with bait alone the response was coordinated by the national Consultative Committee on Emergency Plant Pests (CCEPP) guided by a National Biosecurity Consultative Committee (NBCC) including entomologists. These specialists would have had experience from the other ant inclusions. The authors cite none of this literature The CCEPP and NBCC are mentioned in the manuscript. The study seemed to downplay the methods involved in treating the ant population (assumed it would be easy) and seemed to have assumed the ants could be eradicated with fipronil baits. Most entomologists in this area, and the published literature, would have likely indicated that this was not likely possible. The tenor of the study seemed to go about assuming ant eradication and was fixed mostly on the models to prove it. Yes, this is a manuscript describing the statistic methods to support the claim for eradication. The details of the eradication methods were not overly detailed as the focus of this manuscript was on the statistical methods.          

The history of eradication in the field of entomology is very poor. I can think of one case, and annual monitoring (decades after this insect was deemed "eradicated") continues for this insect because entomologists know that eradication is unlikely the purpose of the statistical methods is to provide managers with confidence that eradication is likely to be successful. It is designed to support the decision to discontinue needless surveillance. And in fact, the YCA was not eradicated from Lismore eradication was accepted by the national CCEPP so we authors and the national body disagree with the reviewer’s view. Perhaps for better clarity, we specified the period covering this epidemiological study. On lines 339-341 the authors state that the ants were eradicated as accepted by the national CCEPP, but then in Feb 2021 (lines 356-358) the ants returned yes, they were found again however it is unclear if this was an undetected population for two years or a reintroduction. Therefore, the ants were not eradicated the national CCEPP disagrees with this view. The word eradication should not be used in this MS, especially in the title we disagree because this was a national agreement – rediscovery could have been by reintroduction.

The authors provide no evidence of the efficacy of dogs in finding ants our focus was on the statists to support claims for eradication. Therefore, we provided an overview of the methods as a background. The methodology seems appealing, so it seems the authors just assumed the dogs were efficient surveillance dogs are used routinely in airports – we have all seen them – this technique is not new. There are many papers of the use of surveillance dogs in many areas of biosecurity ,and even human health. Published studies on the use of dogs for bed bug detection showed wide variability, from dogs that were good at it to dogs that were nearly useless the same could be said about pesticide for ant control. The efficacy of sniffer dogs is based on their training. There's no evidence that dogs were effective at finding the YCAs - only an assumption yes, this was a method supported by the NBCC and CCEPP (both national bodies) so perhaps there is some confidence that the method has merit. This is a statistical epidemiology paper, not a proof of concept for sniffer dogs.

There's no discussion or definition of trapping, luring, and visual these seem self-evident for most readers however I have provided some definitions to make this clearer. The authors use the word "bait trap" this is a term used in official documents – we saw no reason to change the official text. This seems to be verbage suggesting the authors are not well-versed in pest control. The product is a bait incorrect. A bait is something left out and not reinspected. The bait traps used bait formulations and were revisited and assess ant presence and bait consumption. Again, the “bait trap” is the words used in eradication documents. I could not tell whether any sprays, containing fipronil, were used no sprays were used – our manuscript never mentions sprays. The project would have benefited from a spray program not according to the NBCC and CCEPP who provided funds for the eradication and who approval the eradication plans. Bait alone is not as likley to control large ant populations as is the combination - even spot treatments (mls of 0.06% fipronil spray) are highly beneficial. I couldn't tell the percantage of active ingredient in the bait text changed to better explain the formulation. There was little literature, if any, cited on the biology of the YCA or on ant managment with fipronil this is an epidemiology manuscript, not a world-wide review of all things YCA. Besides, much of that was summarized in Dominiak et al. (2011). A brief lit review of the ant would have suggested that they are honeydew tenders, and therefore prefer sweet liquid baits the program was governed by the NBCC and CCEPP. 

Response to hand written notes (where readable) are provided below. For ease of response, I assigned a number to each handwritten note and my response is based on that number. There are many text passages circled by no edits offered and I assumed these were for the reviewer’s benefit. I did not alter those circled text passages.

Responses

  1. We redefined the period covering the eradication. The reviewer may consider that eradication was not successful but the national NBCC and CCEPP did accept the statistical claim for eradication. The subsequent detection in February 2021 may have been a reintroduction.
  2. The “no entomologist = no eradication” assumes that only entomologists can make these decisions. The case for eradication was accepted by two national bodies. So, we disagree with this comment.
  3. This is an epidemiology manuscript, describing the statistical case/analysis to demonstrate freedom. We redefined the study period to cover the period to the point where national bodies agreed to the case for eradication.
  4. Text amended as suggested.
  5. Text amended as suggested.
  6. This appears to be a comment to the reviewer but no correction offered. Text not changed.
  7. Text amended as suggested.
  8. Text amended as suggested.
  9. We think your text is more informative. Text not changed.
  10. I can’t read the handwriting.
  11. In English in this sentence, the suggested “it” would refer to the last noun in the previous sentence which was “produce”. So starting the sentence with “it” would infer that “produce had been detected” which is not the case. The use of our text in this sentence is a better use of English without confusion. Text not changed.
  12. Similarly, “it” would refer to the landscapes and this is poor English, leading to confusion. For better clarity, “YCA” is better English. Text not changed.
  13. Text amended as suggested.
  14. I disagree but amended the text as suggested.
  15. Text amended as suggested.
  16. There are major endemic and exotic threats to Australia. The use of “exotic” was deliberate. Text not changed.
  17. Text amended as suggested.
  18. Text amended as suggested.
  19. “were” is more direct verb – why use two words when one word will do. Next not changed.
  20. Text not changed. The three words help the reader transition for the previous broad paragraph to the narrow view on YCA. This wording is consistent with the funnel shape of introductions based on our training for scientific writing.
  21. YCA describes a single species therefore the verb should be singular. Text not changed.
  22. Text amended as suggested.
  23. Text amended as suggested.
  24. Technically, YCA looks for any source of sugar. Honeydew is just one example. Technically, our text is more correct. Text amended to include “honeydew”.
  25. Text amended as suggested.
  26. Text circled with no suggested change. This implies the reviewer does not understand the concept. Text not changed.
  27. “Local control centre” is a standard term in emergency response and is self-evident. Given the many insect incursions and their publicity, I would think most people are familiar with emergency responses. I added text for the organising authority for clarity.
  28. Yes, we agree that, based on the level of detections, that this was a late detection and well established. We think most readers would draw the same conclusion therefore we did not amend our text. It is self-evident.
  29. Yes – there was a considerable distance between the two sites. Text not changed as this appears to be a comment, not a correction.
  30. In regard to the timeline, the paragraph above states that the detection was confirmed on 21 May. On Line 123, we state that surveillance started on 21 May. The local control centre was established on 4 June (Line 101). Public consultation began on 29 June (Line 133). Initial baiting started 10 June (Line 161). The timeline already exists in our text. Our text not changed.
  31. Yes, the CBD is a considerable area.
  32. The terms “trapping”, “luring”, “odour detection dogs” and “visual” are common in surveillance and emergency response. Also, they are largely self evident. We did not want to delve into these definitions as this is an epidemiology manuscript, focusing on the statistical approach. However, I provided additional text on lures and visual inspections.
  33. The “problematic” bracket covers 18 lines of text and it is unclear if all or part of the text is “problematic”. Text not changed.
  34. The question of data efficacy is raised regarding the merits of sniffer dogs or odour detection dogs. Dogs are widely used for many surveillance programs and are a broadly accepted technique, even though the reviewer questions their value. We did not add efficacy data as we do not have the data on this widely and commonly used technique. Our text was not changed.
  35. This seems to be the same issue as 34.
  36. Again, the reviewer is concerned that there were no entomologists on the paper. This is an epidemiology paper using ants as an example. Entomologists would have been in the NBCC and advised the CCEPP. So, entomologists would had influence on the program activities. But they did not fulfill the requirements of authorship according to national guidelines. This is an epidemiology paper – not an entomology paper. If it helps, I have a PhD in fruit fly management with 22 years experience – but my base degree was not in entomology.
  37. Active surveillance and passive surveillance are standard surveillance terms – and largely self-evident. Text not changed.
  38. We changed the text on the permit to explain this better.
  39. Text amended as suggested.
  40. We never used sprays – we only used baits. Text not changed.
  41. We changed the text on the permit to explain this better.
  42. We amended the text on the permit to better explain these details.
  43. The term “bait traps” were used in official documents. We saw no reason to change the term. Some explanatory text was added.
  44. All the text refers to Antoff bait. Text not changed.
  45. Goonellabah is another location – we added some text.
  46. There were two permits. The formulation of the fipronil/sugar/water/hydrol gel was explained in new text, raised in a previous change.
  47. Concentrations are provided in the revised text on permits.
  48. This is known because bait traps are revisited and the amount of product removed was visually assessed. This is explained in text raised in a previous point.
  49. The issue about bait and bait trap has been addressed in a previously raised point.
  50. No, sprays were never used. There is no suggestion in our text that sprays were even considered. The section on permits and pesticides only describes baits.
  51. I amended the caption for both figures to make it more clear.
  52. I altered the caption to figure 2 to make the explanation clearer.
  53. Active and passive surveillance are standard surveillance terms and are largely self-explanatory. Text not changed.
  54. I though we explained how the model was developed and used, in our methods. I suspect the reviewer made this notation before reading the next three pages.
  55. I added reference [13] – already in the list of references.
  56. Baiting alone may be “unlikely” to eradicate but it is not impossible. We reported faithfully on our eradication work and oor results are different to the expectations of the reviewer. Perhaps the likelihood of eradication might be based on how the baiting was delivered and for how long. Text not changed as there was no suggested correction.
  57. The details of the pesticide preparation were clarified because of an earlier correction.
  58. I don’t know how much vegetation was in the CBD but most CBDs have vegetation for beautification.
  59. Text amended as suggested.
  60. New reference [27] added and subsequent citation numbers adjusted.
  61. Text amended as suggested.
  62. This is a general statement regarding the detection of exotic incursions. 9 ha is very small (and early) compared to the fire ant detection in Queensland covering 100’s of hectares. Text not changed.
  63. There was no broadcast spray – there was broadcast baiting. Our text never mentions a spray.
  64. Same as point 63.
  65. I appreciate that the record of insect eradication is perceived to be poor by the reviewer, however it is not impossible. Text not changed because this appears to be a comment rather than a correction.
  66. We redefined the period covering the eradication. The reviewer may consider that eradication was not successful but the national bodies of NBCC and CCEPP did accept the statistical case for eradication. The subsequently detection in February 2021 may have been a reintroduction.
  67. I have provided additional text on the difference between bait and trap bait in a response to a previous point.

Reviewer 2 Report

Comments and Suggestions for Authors

I have made edits (grammar) and comments and inquiries (that must be addressed) in pencil on the MS for the author's consideration. The MS is well-written, but in area some things are not explained or defined (methods) or left unmentioned.

The study would have greatly benefited by having an entomologist/myrmecologist involved, esp one experienced in ant management. Ant control is quite difficult, and consultation with an experienced entomologist might have highlighted that fact that large ant populations (tawny crazy ant, Argentine ant) are not eliminted with bait alone. The authors cite none of this literature. The study seemed to downplay the methods involved in treating the ant population (assumed it would be easy) and seemed to have assumed the ants could be eradicated with fipronil baits. Most entomologists in this area, and the published literature, would have likely indicated that this was not likely possible. The tenor of the study seemed to go about assuming ant eradication and was fixed mostly on the models to prove it.

The history of eradication in the field of entomology is very poor. I can think of one case, and annual monitoring (decades after this insect was deemed "eradicated") continues for this insect because entomologists know that eradication is unlikely. And in fact, the YCA was not eradicated from Lismore. On lines 339-341 the authors state that the ants were eradicated, but then in Feb 2021 (lines 356-358) the ants returned. Therefore, the ants were not eradicated. The word eradication should not be used in this MS, especially in the title.

The authors provide no evidence of the efficacy of dogs in finding ants. The methodology seems appealing, so it seems the authors just assumed the dogs were efficient. Published studies on the use of dogs for bed bug detection showed wide variability, from dogs that were good at it to dogs that were nearly useless. There's no evidene that dogs were effective at finding the YCAs - only an assumption.

There's no discussion or definition of trapping, luring, and visual. The authors use the word "bait trap". This seems to be verbage suggesting the authors are not well-versed in pest control. The product is a bait. I could not tell whether any sprays, containing fipronil, were used. The project would have benefited from a spray program. Bait alone is not as likley to control large ant populations as is the combination - even spot treatments (mls of 0.06% fipronil spray) are highly beneficial. I couldn't tell the percantage of active ingredient in the bait. There was little literature, if any, cited on the biology of the YCA or on ant managment with fipronil. A brief lit review of the ant would have suggested that they are honeydew tenders, and therefore prefer sweet liquid baits. 

Author Response

Reviewer 2

I have made edits (grammar) and comments and inquiries (that must be addressed) in pencil on the MS for the author's consideration. Thanks for these comments. I will number each comment and respond where I can read the handwritten. The MS is well-written thank you, but in area some things are not explained or defined (methods) or left unmentioned.

The study would have greatly benefited by having an entomologist/myrmecologist involved, esp one experienced in ant management This is an epidemiology paper using ants as the example. It is not primarily an entomology paper. Ant control is quite difficult, and consultation with an experienced entomologist might have highlighted that fact that large ant populations (tawny crazy ant, Argentine ant) are not eliminated with bait alone the response was coordinated by the national Consultative Committee on Emergency Plant Pests (CCEPP) guided by a National Biosecurity Consultative Committee (NBCC) including entomologists. These specialists would have had experience from the other ant inclusions. The authors cite none of this literature The CCEPP and NBCC are mentioned in the manuscript. The study seemed to downplay the methods involved in treating the ant population (assumed it would be easy) and seemed to have assumed the ants could be eradicated with fipronil baits. Most entomologists in this area, and the published literature, would have likely indicated that this was not likely possible. The tenor of the study seemed to go about assuming ant eradication and was fixed mostly on the models to prove it. Yes, this is a manuscript describing the statistic methods to support the claim for eradication. The details of the eradication methods were not overly detailed as the focus of this manuscript was on the statistical methods.          

The history of eradication in the field of entomology is very poor. I can think of one case, and annual monitoring (decades after this insect was deemed "eradicated") continues for this insect because entomologists know that eradication is unlikely the purpose of the statistical methods is to provide managers with confidence that eradication is likely to be successful. It is designed to support the decision to discontinue needless surveillance. And in fact, the YCA was not eradicated from Lismore eradication was accepted by the national CCEPP so we authors and the national body disagree with the reviewer’s view. Perhaps for better clarity, we specified the period covering this epidemiological study. On lines 339-341 the authors state that the ants were eradicated as accepted by the national CCEPP, but then in Feb 2021 (lines 356-358) the ants returned yes, they were found again however it is unclear if this was an undetected population for two years or a reintroduction. Therefore, the ants were not eradicated the national CCEPP disagrees with this view. The word eradication should not be used in this MS, especially in the title we disagree because this was a national agreement – rediscovery could have been by reintroduction.

The authors provide no evidence of the efficacy of dogs in finding ants our focus was on the statists to support claims for eradication. Therefore, we provided an overview of the methods as a background. The methodology seems appealing, so it seems the authors just assumed the dogs were efficient surveillance dogs are used routinely in airports – we have all seen them – this technique is not new. There are many papers of the use of surveillance dogs in many areas of biosecurity ,and even human health. Published studies on the use of dogs for bed bug detection showed wide variability, from dogs that were good at it to dogs that were nearly useless the same could be said about pesticide for ant control. The efficacy of sniffer dogs is based on their training. There's no evidence that dogs were effective at finding the YCAs - only an assumption yes, this was a method supported by the NBCC and CCEPP (both national bodies) so perhaps there is some confidence that the method has merit. This is a statistical epidemiology paper, not a proof of concept for sniffer dogs.

There's no discussion or definition of trapping, luring, and visual these seem self-evident for most readers however I have provided some definitions to make this clearer. The authors use the word "bait trap" this is a term used in official documents – we saw no reason to change the official text. This seems to be verbage suggesting the authors are not well-versed in pest control. The product is a bait incorrect. A bait is something left out and not reinspected. The bait traps used bait formulations and were revisited and assess ant presence and bait consumption. Again, the “bait trap” is the words used in eradication documents. I could not tell whether any sprays, containing fipronil, were used no sprays were used – our manuscript never mentions sprays. The project would have benefited from a spray program not according to the NBCC and CCEPP who provided funds for the eradication and who approval the eradication plans. Bait alone is not as likley to control large ant populations as is the combination - even spot treatments (mls of 0.06% fipronil spray) are highly beneficial. I couldn't tell the percantage of active ingredient in the bait text changed to better explain the formulation. There was little literature, if any, cited on the biology of the YCA or on ant managment with fipronil this is an epidemiology manuscript, not a world-wide review of all things YCA. Besides, much of that was summarized in Dominiak et al. (2011). A brief lit review of the ant would have suggested that they are honeydew tenders, and therefore prefer sweet liquid baits the program was governed by the NBCC and CCEPP. 

Response to hand written notes (where readable) are provided below. For ease of response, I assigned a number to each handwritten note and my response is based on that number. There are many text passages circled by no edits offered and I assumed these were for the reviewer’s benefit. I did not alter those circled text passages.

Responses

  1. We redefined the period covering the eradication. The reviewer may consider that eradication was not successful but the national NBCC and CCEPP did accept the statistical claim for eradication. The subsequent detection in February 2021 may have been a reintroduction.
  2. The “no entomologist = no eradication” assumes that only entomologists can make these decisions. The case for eradication was accepted by two national bodies. So, we disagree with this comment.
  3. This is an epidemiology manuscript, describing the statistical case/analysis to demonstrate freedom. We redefined the study period to cover the period to the point where national bodies agreed to the case for eradication.
  4. Text amended as suggested.
  5. Text amended as suggested.
  6. This appears to be a comment to the reviewer but no correction offered. Text not changed.
  7. Text amended as suggested.
  8. Text amended as suggested.
  9. We think your text is more informative. Text not changed.
  10. I can’t read the handwriting.
  11. In English in this sentence, the suggested “it” would refer to the last noun in the previous sentence which was “produce”. So starting the sentence with “it” would infer that “produce had been detected” which is not the case. The use of our text in this sentence is a better use of English without confusion. Text not changed.
  12. Similarly, “it” would refer to the landscapes and this is poor English, leading to confusion. For better clarity, “YCA” is better English. Text not changed.
  13. Text amended as suggested.
  14. I disagree but amended the text as suggested.
  15. Text amended as suggested.
  16. There are major endemic and exotic threats to Australia. The use of “exotic” was deliberate. Text not changed.
  17. Text amended as suggested.
  18. Text amended as suggested.
  19. “were” is more direct verb – why use two words when one word will do. Next not changed.
  20. Text not changed. The three words help the reader transition for the previous broad paragraph to the narrow view on YCA. This wording is consistent with the funnel shape of introductions based on our training for scientific writing.
  21. YCA describes a single species therefore the verb should be singular. Text not changed.
  22. Text amended as suggested.
  23. Text amended as suggested.
  24. Technically, YCA looks for any source of sugar. Honeydew is just one example. Technically, our text is more correct. Text amended to include “honeydew”.
  25. Text amended as suggested.
  26. Text circled with no suggested change. This implies the reviewer does not understand the concept. Text not changed.
  27. “Local control centre” is a standard term in emergency response and is self-evident. Given the many insect incursions and their publicity, I would think most people are familiar with emergency responses. I added text for the organising authority for clarity.
  28. Yes, we agree that, based on the level of detections, that this was a late detection and well established. We think most readers would draw the same conclusion therefore we did not amend our text. It is self-evident.
  29. Yes – there was a considerable distance between the two sites. Text not changed as this appears to be a comment, not a correction.
  30. In regard to the timeline, the paragraph above states that the detection was confirmed on 21 May. On Line 123, we state that surveillance started on 21 May. The local control centre was established on 4 June (Line 101). Public consultation began on 29 June (Line 133). Initial baiting started 10 June (Line 161). The timeline already exists in our text. Our text not changed.
  31. Yes, the CBD is a considerable area.
  32. The terms “trapping”, “luring”, “odour detection dogs” and “visual” are common in surveillance and emergency response. Also, they are largely self evident. We did not want to delve into these definitions as this is an epidemiology manuscript, focusing on the statistical approach. However, I provided additional text on lures and visual inspections.
  33. The “problematic” bracket covers 18 lines of text and it is unclear if all or part of the text is “problematic”. Text not changed.
  34. The question of data efficacy is raised regarding the merits of sniffer dogs or odour detection dogs. Dogs are widely used for many surveillance programs and are a broadly accepted technique, even though the reviewer questions their value. We did not add efficacy data as we do not have the data on this widely and commonly used technique. Our text was not changed.
  35. This seems to be the same issue as 34.
  36. Again, the reviewer is concerned that there were no entomologists on the paper. This is an epidemiology paper using ants as an example. Entomologists would have been in the NBCC and advised the CCEPP. So, entomologists would had influence on the program activities. But they did not fulfill the requirements of authorship according to national guidelines. This is an epidemiology paper – not an entomology paper. If it helps, I have a PhD in fruit fly management with 22 years experience – but my base degree was not in entomology.
  37. Active surveillance and passive surveillance are standard surveillance terms – and largely self-evident. Text not changed.
  38. We changed the text on the permit to explain this better.
  39. Text amended as suggested.
  40. We never used sprays – we only used baits. Text not changed.
  41. We changed the text on the permit to explain this better.
  42. We amended the text on the permit to better explain these details.
  43. The term “bait traps” were used in official documents. We saw no reason to change the term. Some explanatory text was added.
  44. All the text refers to Antoff bait. Text not changed.
  45. Goonellabah is another location – we added some text.
  46. There were two permits. The formulation of the fipronil/sugar/water/hydrol gel was explained in new text, raised in a previous change.
  47. Concentrations are provided in the revised text on permits.
  48. This is known because bait traps are revisited and the amount of product removed was visually assessed. This is explained in text raised in a previous point.
  49. The issue about bait and bait trap has been addressed in a previously raised point.
  50. No, sprays were never used. There is no suggestion in our text that sprays were even considered. The section on permits and pesticides only describes baits.
  51. I amended the caption for both figures to make it more clear.
  52. I altered the caption to figure 2 to make the explanation clearer.
  53. Active and passive surveillance are standard surveillance terms and are largely self-explanatory. Text not changed.
  54. I though we explained how the model was developed and used, in our methods. I suspect the reviewer made this notation before reading the next three pages.
  55. I added reference [13] – already in the list of references.
  56. Baiting alone may be “unlikely” to eradicate but it is not impossible. We reported faithfully on our eradication work and oor results are different to the expectations of the reviewer. Perhaps the likelihood of eradication might be based on how the baiting was delivered and for how long. Text not changed as there was no suggested correction.
  57. The details of the pesticide preparation were clarified because of an earlier correction.
  58. I don’t know how much vegetation was in the CBD but most CBDs have vegetation for beautification.
  59. Text amended as suggested.
  60. New reference [27] added and subsequent citation numbers adjusted.
  61. Text amended as suggested.
  62. This is a general statement regarding the detection of exotic incursions. 9 ha is very small (and early) compared to the fire ant detection in Queensland covering 100’s of hectares. Text not changed.
  63. There was no broadcast spray – there was broadcast baiting. Our text never mentions a spray.
  64. Same as point 63.
  65. I appreciate that the record of insect eradication is perceived to be poor by the reviewer, however it is not impossible. Text not changed because this appears to be a comment rather than a correction.
  66. We redefined the period covering the eradication. The reviewer may consider that eradication was not successful but the national bodies of NBCC and CCEPP did accept the statistical case for eradication. The subsequently detection in February 2021 may have been a reintroduction.
  67. I have provided additional text on the difference between bait and trap bait in a response to a previous point.